# Organellar Evolution: A Path from Benefit to Dependence

**DOI:** 10.3390/microorganisms10010122

**Published:** 2022-01-07

**Authors:** Miroslav Oborník

**Affiliations:** 1Institute of Parasitology, Biology Centre, Czech Academy of Sciences, 37005 České Budějovice, Czech Republic; obornik@paru.cas.cz; 2Faculty of Science, University of South Bohemia, 37005 České Budějovice, Czech Republic

**Keywords:** organelle, endosymbiosis, plastid, mitochondrion, benefit, essential function

## Abstract

Eukaryotic organelles supposedly evolved from their bacterial ancestors because of their benefits to host cells. However, organelles are quite often retained, even when the beneficial metabolic pathway is lost, due to something other than the original beneficial function. The organellar function essential for cell survival is, in the end, the result of organellar evolution, particularly losses of redundant metabolic pathways present in both the host and endosymbiont, followed by a gradual distribution of metabolic functions between the organelle and host. Such biological division of metabolic labor leads to mutual dependence of the endosymbiont and host. Changing environmental conditions, such as the gradual shift of an organism from aerobic to anaerobic conditions or light to dark, can make the original benefit useless. Therefore, it can be challenging to deduce the original beneficial function, if there is any, underlying organellar acquisition. However, it is also possible that the organelle is retained because it simply resists being eliminated or digested untill it becomes indispensable.

## 1. Introduction

A eukaryotic cell is typical by hosting semiautonomous organelles, such as mitochondria and plastids. These organelles are deeply integrated into the host cell; however, they usually keep some level of independence by encoding a fraction of the organellar proteome and RNAs in their genomes [1,2,3,4], living to some extent like endosymbiotic bacteria [5,6]. Mitochondria and plastids are, with few exceptions, essential for the host cell survival; once the cell has captured an organelle, it can hardly get rid of it [1,2,3,4,6,7]. It is believed that mitochondria and plastids evolved in endosymbiotic events, involving an engulfment or invasion of a free-living organellar ancestor, followed by the endosymbiotic transfer of genes from the captured entity to the nucleus of the host cell, with a consequent import of nuclear-encoded proteins into the organelle [3,8,9]. Symbiosis is an intimate, long-time relationship of two dissimilar organisms living together [10]. Although it is often understood as mutualism, the state beneficial for both partners, symbiosis, in fact, involves a continuum of relationships ranging from mutualism to parasitism [11]. The evolutionary history of plastids by domesticating a cyanobacterium is apparent because they are evolutionarily younger, and a cyanobacterial ancestor was likely acquired by the regular eukaryotic cell capable of phagocytosis [3,8,9].

On the other hand, the origin of the evolutionary older mitochondrion is more elusive. It is not straightfoward as to what kind of cell engulfed the mitochondrial ancestor, what ancestor it was, what the original mitochondrial beneficial function was, if it had any, and what kind of a symbiotic relationship the endosymbiotic partners had [6,12,13,14]. The veil of time successfully obscures the evolutionary history of the mitochondrion.

## 2. The Plastid Benefit for the Host Is Photosynthesis

When talking about organelles, it is believed that benefits drive symbiotic relationships. However, it can be pretty challenging to trace the original benefit for which the endosymbiont is retained and integrated into the host cell. It is most likely that photosynthesis was the reason for adopting a cyanobacterial symbiont by a heterotrophic eukaryotic host (Figure 1) because there is no other way to get photosynthesis into the eukaryotic cell lacking it. Photosynthesis is managed by such complex molecular machinery that convergent evolution of the machinery appears highly improbable [6]. While the lateral gene transfer (LGT) likely stands behind the evolution of photosynthesis in prokaryotes [15,16,17], it played, except the endosymbiotic LGT, no role in the evolution of photosynthesis in eukaryotes. Instead, the evolutionary history of eukaryotic phototrophs is full of endosymbiotic events involving prokaryotic (at least twice) or eukaryotic phototrophs (many times) as donors of photosynthetic ability. Consequently, compartmentalization always physically separates photosynthesis from the host cell in eukaryotes (Figure 1). Photosynthesis has been transmitted throughout the tree of life for the apparent reason of the acquisition of a photoautotrophic lifestyle. Although it was supposed for a long time that plastid endosymbiotic events are rare in evolution [18,19], it recently appeared that at least two and six independent events were likely responsible for the appearance of primary and complex plastids, respectively, not counting complex plastids replacements. In addition to the primary endosymbiotic Archaeplastida, a relatively recent event involving heterotrophic amoebae and a cyanobacterium was proposed for the rhizarian *Paulinella chromatophora*, again with the apparent benefit of photoautotrophy [20,21,22,23,24]. Complex plastids have likely been independently acquired in Euglenophyta, Chorarachniophyta, Alveolata, Stramenopila, Haptophyta, and Cryptophyta [25,26,27,28,29,30,31] (Table 1). 

Photosynthesis as a beneficial function is, however, not essential for the host cell‘s survival. Many photoautotrophic lineages became secondarily heterotrophic (Figure 1 and Table 1). Various eukaryotes have lost photosynthesis, being either forced by the lack of access to light or by chemical (e.g., antibiotic) disruption of the photosynthetic molecular machine [11,31,44,45,46,47,48,49,50]. For example, apicomplexan parasites (*Sporozoa*, *Apicomplexa*) likely became secondarily heterotrophic because of the easy availability of the organic carbon from the host or by the switch of the ancestral photoparasite from translucent to the opaque host [11]. Additionally, we have to consider that photosynthesis is not only beneficial for the primary producer, as it is quite costly and stands behind the outstanding production of reactive oxygen species (ROS), which can heavily damage the cell [45]. Additionally, a strictly phototrophic lifestyle forces the organisms to live in the access to light and thus limits their environmental distribution.

**Figure 1 microorganisms-10-00122-f001:**
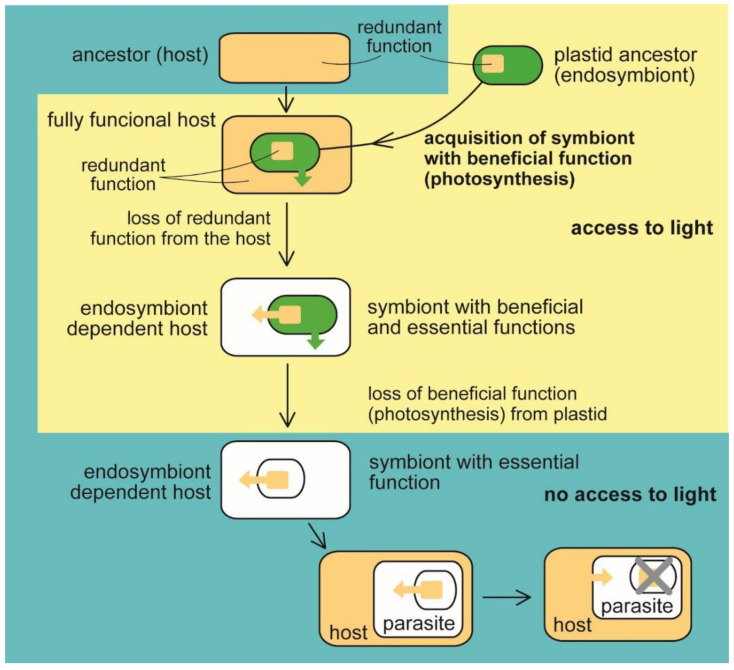
Evolution of benefit and essential function in the plastid. The heterotrophic host acquired a photosynthetic endosymbiotic bacterium with the function (photosynthesis) beneficial for the host. The host cell lost the redundant function (e.g., synthesis of heme, fatty acids, and isoprenoids). At the same time, the delegation of the syntheses to the endosymbiont makes it essential for host survival (eukaryotic phototrophs, e.g., Archaeplastida and *Paulinella* sp., and algae with complex plastids such as Ochrophyta, Cryptophyta, Haptophyta, Dinophyta, Apicomplexa, Euglenopyhta, Chlorarachniophyta [1,2,3,11]). The endosymbiont retained its indispensability for the host even when it had lost photosynthesis, the original beneficial function (in nonphotosynthetic algae, e.g., *Helicosporidium* sp., *Polytomella* sp., *Euglena longa*, apicomplexan parasites, for example, *Plasmodium falciparum*, *Toxoplasma gondii* [1,2,3,7,11] Table 1). Switching to parasitism and scavenging the essential compounds from the host allows the complete loss of the plastid (apicomplexan parasite *Cryptosporidium* [42], parasitic dinoflagellate *Hematodinium* [40]).

Therefore, many phototrophs are, in fact, mixotrophs, which can still live heterotrophically. Such organisms may be prone to losing photosynthesis when getting to the nutrient-rich environment or finding a successful heterotrophic strategy, such as predation or parasitism. Moreover, many protists combine the phototrophic and heterotrophic lives to overcome a reduced availability of various compounds present in host or prey but rare in their environment, such as, for example, nitrogen, phosphorus, iron, and sulfur [11]. Therefore, photosynthesis was frequently lost from green parasitic algae [31,44], euglenophytes [46], apicomplexan parasites [11,31,47,48,49,50], and dinoflagellates [48]. Various secondarily heterotrophic strategies, including parasitism, have evolved repeatedly during the evolution of life among former phototrophs [11,38,39]. It is worth noting that such trophic switches are often found in the same taxonomic groups, such as, for example, in myozozans, group of alveolate protists, consisting of dinoflagellates, apicomplexan parasites, and apicomonads (chromerids and colpodellids). Apicomplexans and dinoflagellates are also the only known algae shown to lose their plastids completely. The plastid is absent from the apicomplexan parasitic genus *Cryptosporidium* [42] and gregarines [51], and the parasitic dinoflagellates of the genus *Hematodinium* [40]. The plastid losses have happened exclusively in parasites thanks to their ability to scavenge the essential compounds originally produced by plastids from the host (Table 2).

## 3. A Beneficial Function of the Mitochondrion

In contrast with plastids, the hypothetical benefit responsible for acquiring mitochondria is the subject of speculation. Frankly speaking, it is not even clear if the mitochondrion was indispensable for forming an early eukaryotic cell [12] or when it was acquired in the course of evolution (early versus late acquisition). The discovery of the eukaryote lacking a mitochondrion proved that the organelle is not essential for the eukaryotic cell as it exists now when the organism is a secondary anaerobic (Figure 2) [61]. Generally, mitochondria are great examples of reductive evolution. The diversity of this organelle involves mitochondria with large circular genomes (e.g., in jakobids [52,53,62]), mitochondria with highly reduced genomes (e.g., those in apicomplexans; [63]), mitochondria-related organelles (MRO), e.g., mitosomes and hydrogenosomes without genomes [60], and, in the end, completely lost mitochondrion (in oxymonads; Table 2 [61]). Consequently, various such organelles possess diverse molecular machinery: the complete respiratory chain of aerobic mitochondria or variously reduced respiratory chains lacking particular complexes: complex I (myzozans, e.g., apicomplexan parasites, and some fungi), complexes III and IV, while complexes I, II, and V retained [64], complexes I and III in chromerids [58,63] and the dinoflagellates of the genus *Amoebophrya* [59]), or complexes III, IV, and ATP synthase in (some) hydrogenosomes [60]. The missing complexes are substituted by alternative sources of electrons (e.g., alternative NADH dehydrogenases, L- and D- lactate cytochrome *c* oxidoreductases), or the electron transport chain was even completely lost (MRO). 

Mitochondria host a wide range of metabolic functions, with oxidative phosphorylation being the most prominent (Figure 2), although it is found only in classical mitochondrial organelles, and lost from highly reduced MROs. In addition to that, mitochondria can be responsible for the metabolism of amino acids and nucleotides, steroid biosynthesis, heme synthesis, fatty acid catabolism, iron-cluster biogenesis, and many others [60,65,66,67,68]. Such metabolic complexity makes the search for the original beneficial function of the mitochondrion difficult. Various hypotheses have been formulated to explain the primary reason for acquiring a mitochondrion. It is further complicated by extensive genetic rearrangements of the organelle during organellogenesis because a substantial part of the mitochondrial proteome does not originate from the supposed mitochondrial ancestor [60].

The hydrogen hypothesis [69] proposed that the primary benefit of pre-mitochondrial symbiont was hydrogen production for the host, methanogenic Archaea. Some eukaryotes, such as *Acanthamoeba castellanii* (*Amoebozoa*), *Brevimastigomonas motovehiculus* (Rhizaria), *Blastocystis* spp. (*Stramenopila*), *Nyctotherus ovalis* (*Alveolata*) still contain hydrogen-producing mitochondria with complete (*Acanthamoeba*) or reduced respiratory chains [60,64,70]. Others host hydrogenosomes, organelles believed to represent modified hydrogen-producing mitochondria [13,60], which have lost the ATP generating part of the respiratory chain and retain just complex I (NADH hydrogenase), or complex II (succinate dehydrogenase) or both [56]. Martin et al. [13] also claim that the mitochondrial ancestor was a facultatively anaerobic bacterium. 

Another possible beneficial function of a pre-mitochondrion was proposed by Thomas Cavalier-Smith [71], who speculated that the ancestor of mitochondrion was a photosynthetic purple bacterium, and the primary benefit was photoautotrophy, similar to plastids. This hypothesis assumes that both symbiont and host were facultative aerobes and that the host already has oxidative phosphorylation. A phototrophic symbiont would have an immediate intracellular synergy between a photosynthetic symbiont fixing CO_2_ and respiring and a phagotrophic host using oxygen and excreting CO_2_ [71].

Other hypotheses suppose that the primary benefit of the mitochondrion is related to dealing with free oxygen in the environment, preferring aerobic heterotrophic respiring bacterium as a mitochondrial ancestor [60]. Oxygen appeared in higher levels during the Great Oxidation Event between 2.4 and 2.1 Bya (billion years ago) [72]. If we take into account the possibility of early acquisition of mitochondrion, the oldest estimates of its appearance touch 2.1 Bya (1655–2094 Mya), while the youngest move around 1 Bya (943–1102 Mya) [73]. One of the earliest views on mitochondrial evolution supposes that the mitochondrion-free anaerobic eukaryotic ancestor acquired aerobic mitochondrion to detoxify oxygen accumulated in the environment after the Great Oxidation Event [74,75]. This scenario would even fit the timing of appearance of eukaryotes between 1 and 2 Bya. However, geochemical data indicate relatively low oxygen levels during the diversification of eukaryotes [72,76]. Moreover, Zimorski et al. [76] argued that those are metabolic processes in the mitochondrion, particularly electron transport chain generating reactive oxygen species, which may harm the cell and must be detoxicated. Therefore, oxidative phosphorylation can hardly be the primary benefit, at least in light of the oxygen detoxification hypothesis. However, we cannot ignore the fact that there is no eukaryote without classical mitochondrion with oxidative phosphorylation known that can permanently live in the presence of oxygen. The use of enzymes that can react with free oxygen does not allow obligate anaerobes to inhabit an oxygen environment. When looking at strictly anaerobic bacteria, they rely on low-potential flavoproteins used for anaerobic respiration. Exposure to oxygen likely causes superoxide and hydrogen peroxide production. They inactivate enzymes with these functional groups through the oxidation of dehydratase iron-sulfur clusters and sulphydryls. However, anaerobes utilize several classes of dioxygen-sensitive enzymes absent from aerobes, which maintain the redox balance during anaerobic fermentation. Their reaction mechanisms require exposure of the solvent of radicals or low-potential metal clusters to the oxygen that can react with it. Additionally, hydroxyl radicals damaging DNA and other biomolecules are generated because hydrogen peroxide oxidizes free iron [77]. Analogously, we can expect that oxygen could not be tolerated by anaerobes involved as the host cell in the early eukaryotic endosymbiotic events. A mitochondrial ancestor could eventually invade the host cell as an energy parasite. The proposed intracellularly parasitic ancestor of the mitochondrion was predicted to bear the ATP/ADP translocase importing ATP from the host. In such a case, there was no beneficial function behind a selection of an endosymbiont (parasite) in the event because the benefit was provided by the host cell [78].

## 4. Accumulation of Benefits

The origin of semiautonomous organelles cannot be seen in a static model. It is an evolutionary dynamic system in which an organelle is not in the cell forever; they can be replaced and lost, leaving leftovers in the genome and metabolism or can even disappear without a trace. Learning from the complex plastids, we see that the plastid replacing the old one partially uses the metabolic equipment of the former cell tenant [68,79]. This mechanism forms a highly evolutionary mosaic metabolic net, particularly in organisms passing through serial endosymbiotic events [7,80,81,82,83]. When applied to the origin of the mitochondrion (Figure 3), such hypothetical serial events can explain the evolutionary highly variable mitochondrial proteomes, where only a minor part of it fits, evolutionarily speaking, to the supposed mitochondrial ancestor [60].

Since every symbiont in the series could have a different beneficial function, it is rather challenging to specify the original benefit of current mitochondria. Even more, some of the symbionts (maybe most of them) may have been parasites [10], with no benefit for the host at the time of the invasion. There could also be theoretically more than a single symbiont present in the host cell at a time because multiple endosymbioses stimulate the evolution of what we see as a mutualistic partnership [6].

## 5. Benefit and Essential Function

Although the beneficial function such as photosynthesis is not necessarily essential for the host cell survival in general, it can be crucial in certain conditions or indispensable for life on a global scale. Thus, for example, an organism living in the absence of organic substances in the environment needs photosynthesis serving as the source of organic carbon and thus became essential for its survival. Although lithotrophic bacteria can also fix CO_2_, they likely play a minor role in the global carbon cycle [84]. On the global scale, photosynthesis is essential for life on Earth as we know it because it is the only way that the energy of sunlight can be transformed to the energy of chemical bond utilizable by all living organisms, including heterotrophs. There is no doubt about the fact that, without photosynthesis, life on Earth, if it could exist, would be present on an incomparably smaller scale than it is now. Therefore, the essentiality of the metabolic function always depends on the particular environmental conditions. 

It is even possible that in eukaryotic organellogenesis, we face two fundamentally different endosymbiotic processes. The first type, presumably leading to the evolution of the mitochondrion, may not need the host cell capable of phagocytoses because it can be based on an active symbiont invasion into the host [6,74]. Since many (maybe most) symbiotic bacteria are or were parasitic in their evolutionary history [6,10], they were likely able to attack and penetrate the host cell and live as intracellular parasites. There was no benefit to the host in such a case, but the relationship was driven by an advantage to the symbiont. Such parasitic relationships could evolve to mutual dependence through the defense of the host cell against the parasite followed by organellogenesis. The second type represented by plastids is based on the phagocytic engulfment of a phototroph which is supposed to retain in the host (originally predatory) cell due to the benefit of photoautotrophy (Figure 4). It is even possible that the symbiotic relationship of a plastid was also not necessarily driven by the benefit of phototrophy but by the resistance of symbiont against digestion by the predatory host cell. Through the stage of kleptoplastidy, non-permanent symbionts, which are, in the end, digested by the host, evolved into permanent plastids [85]. They are, however, kind of “gradually digested” anyway by the host using primary metabolites exported from the symbiont; this process is analogous to the domestication of animals by humans, something like milking cows. Nevertheless, even the permanent plastids are in most eukaryotic phototrophs sometimes lyzed by the host cell, allowing mRNA release to the host cytosol, followed by reverse transcription and the endosymbiotic gene transfer to the host nucleus. Thus, the endosymbiotic gene transfer is an ongoing and never-ending process. Therefore, we can identify differences in the organellar gene repertoires even in groups with conserved plastid (plants) [86] or mitochondrial (animals) genomes [87].

Although endosymbiotic organelles are going through a reductive evolution, constructive evolution also plays a role in the formation of mitochondria and plastids. The protein composition of photosystems (PS) is highly reduced in complex plastids, with the highest level of reduction referred to in chromerids. Although protein subunits have been lost from their photosystems, new ones appeared. Thus, in the apicomonad alga *Chromera velia*, two superoxide dismutases are unprecedentedly permanently attached to the PSI. In addition, three novel subunits with no sequence homology in the databases and unknown functions were found in the PSI of *C. velia* [88]. 

While primary plastids of plants are pretty conserved in their structure and genomes [86], complex plastids in algae display quite diverse functions, and they also can be different in the ultrastructure. For example, while many amino acids s (e.g., glutamate/ glutamine, cysteine, lysine, branched-chain, and aromatic amino acids) are synthesized in the diatom plastid, similar to plants [89], chromerids locate their amino acid synthesis dominantly to the cytosol [90]. 

Mitochondria are somewhat evolutionarily frozen in their function and structure in some groups, such as, for example, animals. In protists, they are highly diverse, as shown in Table 2. Despite the dominant reductive evolution of mitochondria, we can also find, similar to plastids, constructively evolving characters in their genomes. For example, kinetoplastids (e.g., *Trypanosoma* and *Leishmania*) evolved an extremely complex mitochondrial RNA-editing system for the extensive posttranscriptional repair of mRNAs to the translatable form [91].

## 6. Organelle or Symbiotic Bacterium?

Two kinds of biological objects of endosymbiotic origins, mitochondria, and plastids, are currently classified as organelles. Despite their deep genetic, cellular, and metabolic integration into the host cell, they mostly still keep some aspects of individual biological entity: they encode part of the proteome in their own DNA, multiply by regular fission, and move inside the host cell. In my opinion, there is no strictly defined border between organelles and endosymbionts. It looks as though there is a continuum between endosymbionts and organelles, and the term “organelle” is just historically defined, with no solid biological relevance [6].

However, there have been many attempts to find the feature discriminating an organelle from a symbiont [4,6]. The revolutionary but widely accepted definition was proposed by Thomas Cavalier-Smith in 1985 [92]: the organelle exported, by endosymbiotic gene transfer (EGT), its genes to the host nucleus and imports their products from the host. The endosymbiotic gene transfer and nuclear-encoded proteins in the organellar metabolism seem to define organelles. Later, Archibald and Keeling expanded the definition of organelles by three aspects: genetic (EGT and targeting of proteins into the organelle), cellular (synchronization with the host’s cell cycle), and metabolic integration of endosymbiotic partners [4,93]. However, they clearly say that their criteria are soft and that the border between organelles and symbionts will “never be completely unambiguous.” Various symbiotic relationships are understandable as possible evolutionary intermediates from symbionts to organelles. Reyes-Prieto and colleagues [94] introduced the term “symbionelle” for symbionts of insects with highly reduced genomes and consequent metabolic integration to the host. Facultative plant-endosymbiotic nitrogen-fixing bacteria (Rhizobia) import peptides encoded in their plant’s symbiotic partners [95]. The aphid’s γ-proteobacterial symbiont *Buchnera* is enclosed in the host-derived membrane vesicles and imports nuclear-encoded proteins via the secretory pathway. However, no gene transfer from the endosymbiont to the insect cell nucleus has been reported [96]. Finally, it was shown by Husník and colleagues [97] that the mealybug’s endosymbiotic β-proteobacterium *Tremblaya princeps* hosting its own γ-proteobacterial endosymbiont *Moranella endobia*, is supported by 22 nuclear-encoded proteins originating from various bacteria, except for *Tremblaya*. Many of the above-mentioned symbiotic bacteria more or less meet the definition of organelles. However, some of them still keep a higher level of autonomy and do not integrate more with the host, likely because they would lose the benefit of possible transmission into new hosts or a new generation.

On the other hand, the photosynthetic dinoflagellates called “dinotoms” [68,98,99,100,101] host diatom tertiary endosymbionts responsible for their phototrophic lifestyle. These symbionts function as regular plastids; however, they are almost complete diatom cells containing the nucleus, mitochondrion, and the four membraned diatom chloroplasts. The symbiont is separated from the host cell just by a single membrane. Likely due to the single membrane bounding, the symbiont cannot import nuclear-encoded proteins, which may stand behind its unprecedented evolutionary conservation.

The diversity of endosymbiotic relationships in nature is marvelous. It comprises, for example, symbiotic assemblies from facultative nitrogen-fixing symbionts of diatoms and haptophytes [98,99], through nitrogen-fixing bacteria in plants, endosymbionts of insects, bacterial symbionts of various protists [94,95,96,97,98,99,100,101,102,103], and diverse forms of mitochondria [56], to cyanelles of *Paulinella chromatophora* [36,96], and primary and complex plastids of eukaryotic phototrophs and secondary heterotrophs [1,2,3,7,11,30].

## 7. Conclusions

It is believed that the evolution of endosymbiotic organelles is driven by a benefit to the endosymbiotic partners. While the benefit of photosynthesis seems to be obvious in the case of plastids, the original beneficial function of mitochondrion is the subject of discussion. The mosaic evolutionary origin of mitochondrial proteome opens a possibility of serial endosymbiotic events behind the current mitochondria, analogous to that of complex plastids. Plastids and mitochondria can represent two different types of endosymbiotic events: invasion of the alphaproteobacterial intracellular parasite into the pre-eukaryotic host and predation on the cyanobacterial ancestor of plastids. In the mitochondrial evolution, the host has adapted to the presence of the parasite, with a gradual transition of the parasitic symbiosis to a mutualism. Plastids evolved from cyanobacterial prey resistant to digestion by the host cell. In both these models, a benefit for the host was not a driving force for organellogenesis.

## Figures and Tables

**Figure 2 microorganisms-10-00122-f002:**
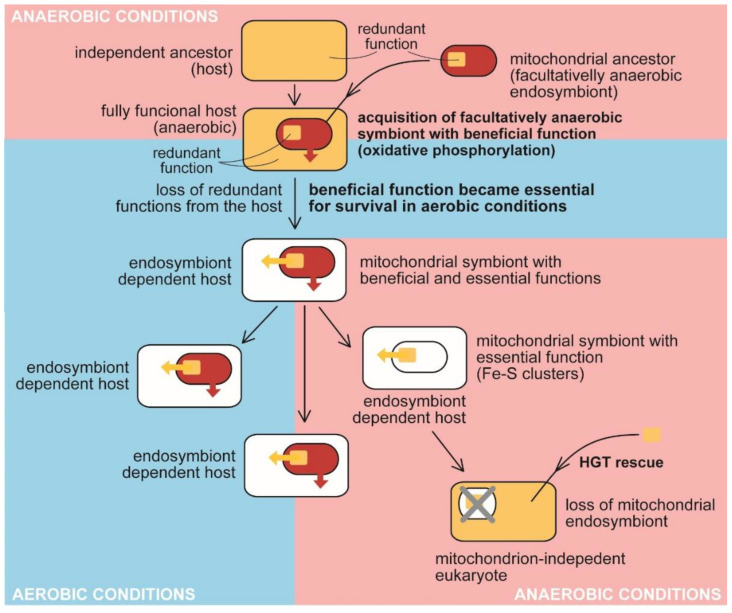
The traditional view on the evolution of benefit and essential function in the mitochondrion. Anaerobic host acquired a facultatively anaerobic endosymbiotic bacterium with the function beneficial for the host, presumably detoxifying oxygen. It became essential for the host in aerobic conditions. The redundant function was lost from the host (e.g., synthesis and assembly of Fe-S clusters). At the same time, the delegation of the synthesis to the endosymbiont makes it essential for host survival. The endosymbiont retained its indispensability for the host even when it had lost the original beneficial function by adaptating to anaerobic conditions. The acquisition of bacterial Fe-S clusters synthesis and assembly in the cytosol of oxymonads through HGT allowed the loss of the mitochondrion (MRO) [61].

**Figure 3 microorganisms-10-00122-f003:**
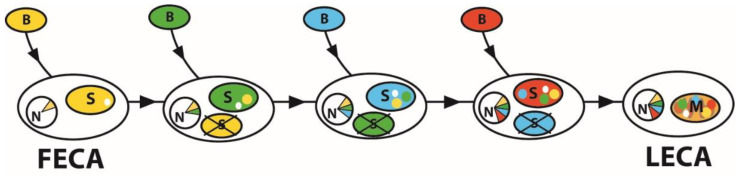
Accumulation of benefits by serial primary endosymbiosis. Each endosymbiont in the evolutionary history from the First Eukaryotic Common Ancestor (FECA) to the Last Eukaryotic Common Ancestor (LECA) impacted the eukaryotic host genome and the proteome of an organelle. The transferred genes (EGT) are supposed to be expressed in the cytosol, and their products are targeted to the last survival, the mitochondrion. It led to the highly chimeric proteomes of the current mitochondria. Partners in the process are bacteria (B), transformed into symbionts (S). The symbiont genes migrate to the host nucleus (N). Such serial events could theoretically lead to the appearance of a mitochondrion (M) with a mosaic proteome.

**Figure 4 microorganisms-10-00122-f004:**
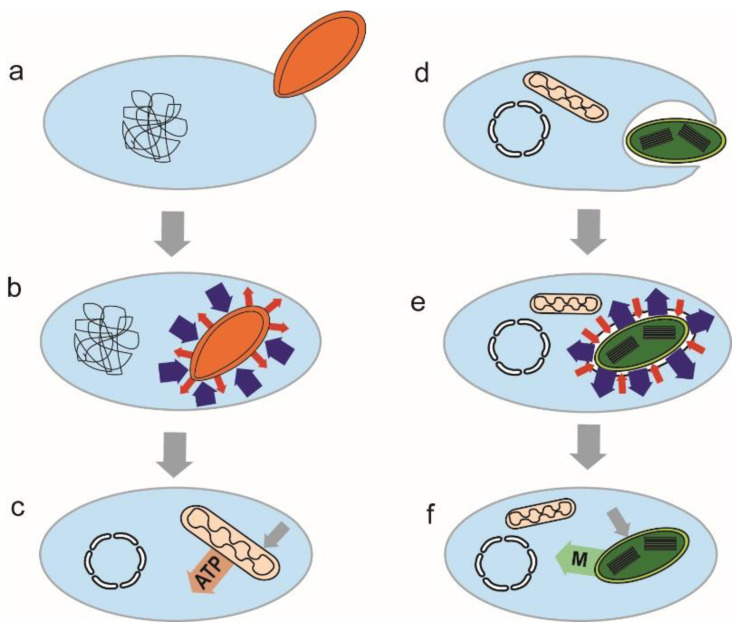
Two possible types of endosymbiotic events. In the first type, the endosymbiont (**a**–**c**) is a parasitic bacterium, which actively invades the host cell (**a**). The defense of the host ((**b**), blue arrows) against parasitic endosymbiont (red arrows) transformed the parasitic relationship to mutualism (**c**). Endosymbiotic gene transfer, together with gradual losses of redundant pathways, led to the mutual dependence of the host and symbiont. The second type of event (**d**–**f**) is based on hunting. The cyanobacterial prey is engulfed by phagocytosis (**d**). The bacterium resists (blue arrows) digestion (red arrows) (**e**). The photosynthetic endosymbiont provides primary metabolites (M) to the host cell in a mutualistic relationship. This type of organellogenesis resembles domestication. In both cases, the endosymbiont provides (non-essential) benefit to the host.

**Table 1 microorganisms-10-00122-t001:** Selected plastids and their characteristics in various eukaryotes. It demonstrates the reductive evolution of plastids in eukaryotes.

Organism	Supergroup	Type of the Plastid	Genes (Cds)	Genome	Reference
*Arabidopsis thaliana*	Archaeplastida	primary	85	circular	[32]
*Porphiridium purpureum*	Archaeplastida	primary	224	circular	[33]
*Helicosporidium* sp.	Archaeplastida	primary	26	circular	[34]
*Polytomella* sp.	Archaeplastida	primary	0	circular	[35]
*Paulinella chromatohpora*	Cercozoa (SAR)	primary (cyanelle)	867	circular	[36]
*Euglena gracilis*	Eugenophyta	complex (secondary)	67	circular	[37]
*Euglena longa*	Eugenophyta	complex (secondary)	46	circular	[38]
*Heterocapsa triquetra*	Dinophyta (SAR)	complex	14	Circular (minicircles)	[39]
*Hematodinium* sp.	Dinophyta (SAR)	-	-	-	[40]
*Thalassiosira pseudonana*	Bacillariophyta (SAR)	complex	141	circular	[41]
*Chromera velia*	Apicomonada (SAR)	complex	78	linear	[42]
*Vitrella brassicaformis*	Apicomonada (SAR)	complex	94	circular	[42]
*Toxoplasma gondii*	Sporozoa (SAR)	complex	29	circular	NCBI
*Cryptosoridium muris*	Sporozoa (SAR)	-	-	-	[43]

**Table 2 microorganisms-10-00122-t002:** Examples of mitochondria and MROs and their characteristics in various eukaryotes. It demonstrates the reductive evolution of mitochondria, from mitochondrial organelles with large genomes to hydrogenosomes and mitosomes lacking genome and eukaryotic cells without mitochondrion.

Species	Taxonomy	Type of Mitochondrion	Genes (Cds)	Genome	Reference
*Andalucia godoyi*	Jakobida	Aerobic/OXPHOS	67	circular	[52]
*Reclinomonas americana*	Jakobida	Aerobic/OXPHOS	66	circular	[53]
*Homo sapiens*	Metazoa (Obazoa)	Aerobic/OXPHOS	13	circular	[54]
*Nymphaea colorata*	Archaeplastida	Aerobic/OXPHOS	42	circular	[55]
*Nyctotherus ovalis*	Ciliophora (SAR)	Anaerobic/H-producing	16	linear	[56]
*Plasmodium falciparum*	Sporozoa (SAR)	Aerobic/OXPHOS	3	linear	[57]
*Chromera velia*	Apicomonada (SAR)	Aerobic/OXPHOS	2	linear	[58]
*Amebophrya ceratii*	Dinophyta (SAR)	Aerobic/OXPHOS	0	-	[59]
*Neocallimastix* sp.	Chytridiomycota (Obazoa)	Hydrogenosome	-	-	[60]
*Giardia intestinalis*	Metamonada	Mitosome (Fe-S clusters)	-	-	[60]
*Monocercomonoides* sp.	Oxymonadida	-	-	-	[61]

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
