# Peer review of "Organellar Evolution: A Path from Benefit to Dependence"

_microorganisms, 2022, doi:10.3390/microorganisms10010122_

Round 1
Reviewer 1 Report
SUMMARY
=======
This is a review of the manuscript entitled "Organellar evolution: a path from benefit to dependence" (microorganisms-1487550) by Oborník. In his manuscript, the author reviewed how free-living bacteria could have given birth to bacterium-derived organelles and why such organelles can be retained despite the change in lifetstyle of the host and, therefore, the absence of any benefits from the organelle. The case of plastid and mitochondrion is detailed from the acquisition to the retention despite the primary benefit to the elimination (for the plastid). Hypothesis about mosaic origin of the mitochondrion is reviewed. The potential benefit or absence of it while establishing the endosymbiosis is also detailed.
The review reads overall well. While I am not expert in organelle evolution, I have some expertise regarding endosymbiosis and parasitism and, therefore, I have some comments listed below.
LEGEND
======
* l.: line
This review is written in [markdown](https://en.wikipedia.org/wiki/Markdown).
MAJOR COMMENTS
==============
My main comments are the following:
* **Symbiosis**: I was happy to read that the author did not associate symbiosis to mutualism (which can often be seen in plant field for instance) but rather to the spectrum from parasitism to mutualism. However, I would recommend defining symbiosis in the introduction because some people may not be familiar with such definition.
* **Parasitic/pathogenic**: "parasitic" and "pathogenic" words were several times associated in the text. While parasites are pathogens (which will manifest by a reduced lifespan at least), the opposite is not true (not all pathogens are parasites). Parasitism is defined by a long-term interaction between host and symbiont. For instance intra-cellular *Listeria* will only interact for some hours with host cells while *Plasmodium* parasites will interact for days (or even years). For the sake of clarity, I would recommend to define parasitism, maybe while defining symbiosis.
Comments on the text
--------------------
* **l. 8**: "Endosymbiotic organelles [...] to the host cell.": This sentence sounds like an abrupt shortcut which can be misleading for people outside the field. Organelles are not entities that are acquired but rather that evolved from bacteria to ultimately become organelles that bring benefit to the host cell. I would suggest rephrasing this sentence in this way.
MINOR COMMENTS
==============
Comments on the text
--------------------
* **l. 284**: extra ".".
Author Response
Replies to reviewer 1
* **Symbiosis**: I was happy to read that the author did not associate symbiosis to mutualism (which can often be seen in plant field for instance) but rather to the spectrum from parasitism to mutualism. However, I would recommend defining symbiosis in the introduction because some people may not be familiar with such definition.
* **Parasitic/pathogenic**: "parasitic" and "pathogenic" words were several times associated in the text. While parasites are pathogens (which will manifest by a reduced lifespan at least), the opposite is not true (not all pathogens are parasites). Parasitism is defined by a long-term interaction between host and symbiont. For instance intra-cellular *Listeria* will only interact for some hours with host cells while *Plasmodium* parasites will interact for days (or even years). For the sake of clarity, I would recommend to define parasitism, maybe while defining symbiosis.
I added into the introduction: Symbiosis is an intimate, long time relationship of two dissimilar organisms living together [10]. Although it is often understood as mutualism, the state beneficial for both partners, symbiosis, in fact, involves a continuum of relationships ranging from mutualism to parasitism [11].
I also deleted „pathogenic“ from the line 257 and legends of figure 4
Comments on the text
--------------------
* **l. 8**: "Endosymbiotic organelles [...] to the host cell.": This sentence sounds like an abrupt shortcut which can be misleading for people outside the field. Organelles are not entities that are acquired but rather that evolved from bacteria to ultimately become organelles that bring benefit to the host cell. I would suggest rephrasing this sentence in this way.
I rephrased the sentence in the abstract:
Eukaryotic organelles supposedly evolved from their bacterial ancestors due to their benefit to the host cell.
MINOR COMMENTS
==============
Comments on the text
--------------------
* **l. 284**: extra ".". corrected
Reviewer 2 Report
This is an interesting review/essay-type paper on the evolution of organelles in diverse eukaryotes. The area is a bit out of my domain but not totally. The overarching theme is that organellar evolution is dynamic and includes phenomenon such as loss of the original beneficial function of the organelle for the host, genetic transfer of genes and other genetic information between different taxa including the host and endosymbiont, and the development of parasitic organelles or hosts parasitizing the organelles. In the review Obornik also covers the idea that mitochondria can have many functions, and the original functions may be unrelated to current benefits. For example, providing H to the host, or detoxifying O2. He also discusses the idea that mitochondria may have originated from photosynthetic symbionts.
Other interesting topics covered include serial endosymbiosis and the accumulation of benefits to the host, and the idea that essential functions sometime occur at a larger scale. For example, all life on Earth largely depends of photosynthesis by green plants. One of the most interesting ideas for me is that organelles may have originated as parasites of the host, providing no benefits. But then in order to protect their own resources they provide resistance to others parasites and pathogens. Overall, he suggests four potential pathways illustrated in Fig. 4.
I think this is a well written and interesting review but I do have a few suggestions:
- Because much of the evolutionary dynamics of organelle evolution has occurred in the unicellular Protists, I think that the paper could be strengthened by including a table or figure that includes species examples and Protist taxa where some of these transitions have occurred.
- Figures 1-4 are interesting and informative but following my first comment, they could be improved by tagging specific steps and transitions to actual organisms with citations. For example, in Figure 2, are there empirical examples of each of these steps that can be documented?
- Finally, following from my comment that most organellar evolution seems to have occurred in Protists, some short discussion is warranted on the historical or current variation in organellar structure or function in existing higher plants and animals today? It seems that all higher plants have essentially the same type of chloroplast. This might be the case with mitochondria in vertebrate animals as well, but I don’t know. My point is that much of organellar evolution has occurred at the prokaryote/protist boundaries and it is not clear how variable the processes and outcomes are in higher eukaryotic taxa.
There were also a couple of small typos on line 150 and 366.
Author Response
Replies to reviewer 2
Because much of the evolutionary dynamics of organelle evolution has occurred in the unicellular Protists, I think that the paper could be strengthened by including a table or figure that includes species examples and Protist taxa where some of these transitions have occurred.
Figures 1-4 are interesting and informative but following my first comment, they could be improved by tagging specific steps and transitions to actual organisms with citations. For example, in Figure 2, are there empirical examples of each of these steps that can be documented?
I modified legend to Fig,1:
Figure 1. Figure 1. Evolution of benefit and essential function in the plastid. The heterotrophic host acquired a photosynthetic endosymbiotic bacteria with the function (photosynthesis) beneficial for the host. The host cell lost the redundant function (e.g., synthesis of heme, fatty acids, and isoprenoids). At the same time, the delegation of the syntheses to the endosymbiont makes it essential for host survival (eukaryotic phototrophs, e.g., Archaeplastida and Paulinella sp., and algae with complex plastids such as Ochrophyta, Cryptophyta, Haptophyta, Dinophyta, Apicomplexa, Euglenopyhta, Chlorarachniophyta [1-3,11]). The endosymbiont retained its indispensability for the host even when it had lost photosynthesis, the original beneficial function (in nonphotosynthetic algae e.g., Helicosporidium sp., Polytomella sp., Euglena longa, apicomplexan parasites, for example, Plasmodium falciparum, Toxoplasma gondii [1-3,7,11] Table 1). Switching to parasitism and scavenging the essential compounds from the host allows the complete loss of the plastid (apicomplexan parasite Cryptosporidium [42], parasitic dinoflagellate Hematodinium [40]).
I also added two tables with examples of plastids and mitochondria in eukaryotes, demonstrating a reductive evolution of both organelles:
Table 1. Selected plastids and their characteristics in various eukaryotes. It demonstrates the reductive evolution of plastids in eukaryotes.
|
organism |
supergroup |
Type of the plastid |
Genes (CDS) |
genome |
reference |
|
|
|
|
|
|
|
|
Arabidopsis thaliana |
Archaeplastida |
primary |
85 |
circular |
[32] |
|
Porphiridium purpureum |
Archaeplastida |
primary |
257 |
circular |
[33] |
|
Helicosporidium sp. |
Archaeplastida |
primary |
26 |
circular |
[34] |
|
Polytomella sp. |
Archaeplastida |
primary |
0 |
circular |
[35] |
|
Paulinella chromatohpora |
Cercozoa (SAR) |
primary (cyanelle) |
867 |
circular |
[36] |
|
Euglena gracilis |
Eugenophyta |
complex (secondary) |
67 |
circular |
[37] |
|
Euglena longa |
Eugenophyta |
complex (secondary) |
46 |
circular |
[38] |
|
Heterocapsa triquetra |
Dinophyta (SAR) |
complex |
14 |
Circular (minicircles) |
[39] |
|
Hematodinium sp. |
Dinophyta (SAR) |
- |
- |
- |
[40] |
|
Thalassiosira pseudonana |
Bacillariophyta (SAR) |
complex |
141 |
circular |
[41] |
|
Chromera velia |
Apicomonada (SAR) |
complex |
78 |
linear |
[42] |
|
Vitrella brassicaformis |
Apicomonada (SAR) |
complex |
94 |
circular |
[42] |
|
Toxoplasma gondii |
Sporozoa (SAR) |
complex |
29 |
circular |
NCBI |
|
Cryptosoridium muris |
Sporozoa (SAR) |
- |
- |
- |
[43] |
Table2. Examples of mitochondria and MROs and their characteristics in various eukaryotes. It demonstrates the reductive evolution of mitochondria, from mitochondrial organelles with large genomes to hydrogenosomes and mitosomes lacking genome, and eukaryotic cell without mitochondrion.
|
species |
taxonomy |
type of mitochondria |
Genes (CDS) |
genome |
reference |
|
Andalucia godoyi |
Jakobida |
Aerobic/OXPHOS |
67 |
circular |
[65] |
|
Reclinomonas americana |
Jakobida |
Aerobic/OXPHOS |
66 |
circular |
[66] |
|
Homo sapiens |
Metazoa (Obazoa) |
Aerobic/OXPHOS |
13 |
circular |
[67] |
|
Nymphaea colorata |
Archaeplastida |
Aerobic/OXPHOS |
42 |
circular |
[68] |
|
Nyctotherus ovalis |
Ciliophora (SAR) |
Anaerobic/H-producing |
16 |
linear |
[69] |
|
Plasmodium falciparum |
Sporozoa (SAR) |
Aerobic/OXPHOS |
3 |
linear |
[70] |
|
Chromera velia |
Apicomonada (SAR) |
Aerobic/OXPHOS |
2 |
linear |
[60] |
|
Amebophrya ceratii |
Dinophyta (SAR) |
Aerobic/OXPHOS |
0 |
- |
[61] |
|
Neocallimastix sp. |
Chytridiomycota (Obazoa) |
Hydrogenosome |
- |
- |
[56] |
|
Giardia intestinalis |
Metamonada |
Mitosome (Fe-S clusters) |
- |
- |
[56] |
|
Monocercomonoides sp. |
Oxymonadida |
- |
- |
- |
[54] |
Finally, following from my comment that most organellar evolution seems to have occurred in Protists, some short discussion is warranted on the historical or current variation in organellar structure or function in existing higher plants and animals today? It seems that all higher plants have essentially the same type of chloroplast. This might be the case with mitochondria in vertebrate animals as well, but I don’t know. My point is that much of organellar evolution has occurred at the prokaryote/protist boundaries and it is not clear how variable the processes and outcomes are in higher eukaryotic taxa.
Thus, the endosymbiotic gene transfer is an ongoing and never-ending process. Therefore we can identify differences in the organellar gene repertoires even in groups with conserved plastid (plants) [90] or mitochondrial (animals) genomes [91].
Although endosymbiotic organelles are going through a reductive evolution, also constructive evolution plays a role in the formation of mitochondria and plastids. The protein composition of photosystems (PS) is highly reduced in complex plastids, with the highest level of reduction referred to in chromerids. Although protein subunits have been lost from their photosystems, new ones appeared. Thus, in the apicomonad alga Chromera velia, two superoxide dismutases are unprecedentedly permanently attached to the PSI. In addition, three novel subunits with no sequence homology in the databases and unknown functions were found in the PSI of C. velia [92].
While primary plastids of plants are pretty conserved in their structure and genomes [90], complex plastids in algae display quite diverse functions, and they also can be different in the ultrastructure. For example, while many amino acids s (e.g., glutamate/ glutamine, cysteine, lysine, branched-chain, and aromatic amino acids) are synthesized in the diatom plastid, similar to plants [93], chromerids locate their amino acid synthesis dominantly to the cytosol [94].
Mitochondria are kind of evolutionary frozen in their function and structure in some groups, such as, for example, animals. In protists, they are highly diverse, as shown in Table 2. Despite the dominant reductive evolution of mitochondria, we can also find, similar to plastids, constructively evolving characters in their genomes. For example, kinetoplastids (e.g., Trypanosoma and Leishmania) evolved an extremely complex mitochondrial RNA-editing system for the extensive posttranscriptional repair of mRNAs to the translatable form [95].
There were also a couple of small typos on line 150 and 366. corrected